# Influence of Topography on UAV LiDAR-Based LAI Estimation in Subtropical Mountainous Secondary Broadleaf Forests

Yunfei Li [1,2,3], Hongda Zeng [1,2,3,*], Jingfeng Xiong [1,2,3] and Guofang Miao [1,2,3]

1 Institute of Geography, Fujian Normal University, Fuzhou 350007, China
2 Key Laboratory for Humid Subtropical Eco-Geographical Processes of the Ministry of Education, School of Geographical Sciences, Fujian Normal University, Fuzhou 350117, China
3 Fujian Sanming Forest Ecosystem National Observation and Research Station, Sanming 365000, China
* Correspondence: zeng.hd@fjnu.edu.cn

**Abstract:** The leaf area index (LAI) serves as a crucial metric in quantifying the structure and density of vegetation canopies, playing an instrumental role in determining vegetation productivity, nutrient and water utilization, and carbon balance dynamics. In subtropical montane forests, the pronounced spatial heterogeneity combined with undulating terrain introduces significant challenges for the optical remote sensing inversion accuracy of LAI, thereby complicating the process of ground validation data collection. The emergence of UAV LiDAR offers an innovative monitoring methodology for canopy LAI inversion in these terrains. This study assesses the implications of altitudinal variations on the attributes of UAV LiDAR point clouds, such as point density, beam footprint, and off-nadir scan angle, and their subsequent ramifications for LAI estimation accuracy. Our findings underscore that with increased altitude, both the average off-nadir scan angle and point density exhibit an ascending trend, while the beam footprint showcases a distinct negative correlation, with a correlation coefficient (R) reaching 0.7. In contrast to parallel flight paths, LAI estimates derived from intersecting flight paths demonstrate superior precision, denoted by $R^2 = 0.70$, RMSE = 0.75, and bias = 0.42. Notably, LAI estimation discrepancies intensify from upper slope positions to middle positions and further to lower ones, amplifying with the steepness of the gradient. Alterations in point cloud attributes induced by the terrain, particularly the off-nadir scan angle and beam footprint, emerge as critical influencers on the precision of LAI estimations. Strategies encompassing refined flight path intervals or multi-directional point cloud data acquisition are proposed to bolster the accuracy of canopy structural parameter estimations in montane landscapes.

**Keywords:** LAI; UAV LiDAR; montane forests; point cloud attributes; flight paths



## 1. Introduction

The vegetation leaf area index (LAI), defined as one half of the total all-sided leaf area per unit ground surface area [1], stands as one of the most crucial biophysical parameters of vegetation canopy structure, governing biophysical processes in vegetation such as transpiration, photosynthesis, and respiration [2–4]. LAI has found extensive applications across forestry, agriculture, and ecology [5]. Thus, an accurate estimation of LAI is of paramount importance for quantitatively assessing the productivity and carbon sequestration functions of forest ecosystems [6], as well as their responses to climate change [7,8].

Mountainous regions cover approximately 24% of the global land area, and China, being a mountainous country, has mountainous areas accounting for 66.7% of its total land area [9]. Mountain vegetation covers a wide area with diverse species, and the topographical variations result in spatial distribution differences in vegetation types at different altitudes and slopes. Portable canopy measurement instruments, such as the LAI-2000, and hemispherical imaging methods based on a fisheye lens are widely recognized as the primary approaches for obtaining ground-based measurements of LAI. These measurement

methods are only applicable on overcast days or within narrow time windows around dawn and dusk. Additionally, light sensors used to measure canopy light intensity require a spacious area or tower top to synchronously measure the light intensity of both the upper and lower canopies within a relatively short timeframe [10,11]. From this, it can be seen that in conducting spatiotemporal monitoring of the forest canopy, LAI incurs significant labor and time costs to obtain a sample size that meets a specific overall prediction accuracy. Remote sensing technology, with its advantages of multisource and persistent acquisition of representative information at various spatiotemporal scales, provides a powerful means for estimating regional-scale vegetation LAI. However, traditional multispectral remote sensing techniques have limited applicability in estimating LAI due to saturation phenomena in remote sensing vegetation indices signals in areas with high vegetation cover [12,13]. Moreover, in contrast with flat terrain, the multispectral remote sensing inversion accuracy of mountainous LAI is primarily influenced by topographic factors, especially variations in image geometry and spectral characteristics caused by undulating terrain. As remote sensing advances in complex topographic areas, the topographic effects on indirectly measured LAI are becoming increasingly apparent [14]. Many studies suggest that under slopes less than 30°, topographic effects have minimal impact on LAI measurements and can be neglected. However, when slopes exceed 30°, topographic effects become a moderate source of error in the indirect measurement of leaf area index [15,16].

LiDAR (light detection and ranging) serves as an active remote sensing technique, capable of capturing the three-dimensional structural information of vegetation. It has been extensively employed for the inversion of canopy structural parameters such as LAI and canopy cover [17–19]. Among LiDAR types, discrete LiDAR point cloud data, due to its straightforward processing, sees more widespread application [20,21]. Currently, there are two primary methods for LAI inversion based on discrete LiDAR: the first combines ground samples with empirical models based on the statistical characteristics of the discrete point cloud [18], while the second employs a physical model inversion method based on the Beer–Lambert Law [22]. The latter uses the ratio of ground point cloud count (intensity) to the total pulse or total point cloud count (intensity) as an approximation for canopy gap fraction, which is then converted to LAI through the Beer–Lambert Law [22]. However, when conducting LiDAR scans in mountainous forests, the undulating terrain can lead to variations in the observed distance between laser pulses and objects. The flight paths planned by unmanned aerial vehicle (UAV) flight control software often fail to meet the requirements for high and uniform point cloud density. Insufficient point cloud density in certain local areas, typically in low-altitude regions, can compromise the accuracy of extracting terrain and canopy structure parameters [23]. Some researchers, by setting different observation flight heights for UAV LiDAR (i.e., 100 m, 150 m, and 200 m), have observed that as the flight height increases, point density decreases, resulting in a reduction in observed information within the tree canopy and a decline in the accuracy of the digital terrain model [24,25]. The LiDAR scanning angle is also another crucial parameter significantly influencing the quantitative estimation of canopy structural parameters [26]. Traditional airborne LiDAR often employs a scanning angle of 15–20° field of view, where a smaller scanning angle increases the chances of vertically incident beams detecting the ground. However, the narrow angle range also limits coverage width. With the development of low-altitude UAVs, the point cloud density increases, and larger scanning angles help optimize costs. Nevertheless, undulating terrain introduces spatial variability in laser pulse incidence angles. Additionally, the increased distance through the canopy due to the inclined incident beam angle may offset the advantages of a wide scanning angle, leading to significant deviations in canopy parameter estimation results and potentially impacting the robustness of LAI estimation results. Hence, Liu et al. [27] recommended avoiding large minimum incident angles (>23°). Currently, there is still limited research explaining how and why LiDAR metrics vary with incidence angles, and the specific changes in LAI estimation errors at certain angles remain largely unknown. Furthermore, slope conditions may result in inaccuracies in individual tree metrics derived

from LiDAR, potentially distorting the plant area index (PAI) contours and altering the spatial distribution patterns of plant area density. This effect becomes more pronounced with increasing slope [28,29].

In summary, drawing on UAV LiDAR and on-the-ground LAI measurements, this study investigates the impact of mountainous terrain factors (such as slope and slope position) on pivotal discrete LiDAR point cloud attributes (namely point density, off-nadir scan angle, and beam footprint) and its implications for LAI estimation. Additionally, the research examines the potential to enhance LAI estimation accuracy in mountainous forests by optimizing UAV LiDAR flight routes.

## 2. Materials and Methods

### 2.1. Study Area and Field Measurements

The study site is located in the Geshi Castanopsis Nature Reserve in Sanming City, Fujian Province, China (26°11′28″ N, 117°28′10″ E), covering an area of 1101.6 ha. The area experiences a subtropical maritime and continental climate with an annual average temperature of 19.5 °C, annual precipitation of 1546.8 mm, a frost-free period of 300 days, an average relative humidity of 79%, and an average wind speed of 1.6 m/s. The predominant soils in the region are yellow soil and red soil, which have been formed primarily from parent materials such as sandstone and shale. These soils generally exhibit a sandy loam texture with a depth exceeding 1 m and often have a thick humus layer. The primary tree species in the community of the study area include Geshi Castanopsis (*Castanopsis kawakamii* Hay.), Masson's pine (*Pinus massoniana* Lamb.), and evergreen oak (*Schima superba* Gardn. et Champ.). The shrub layer is dominated by species such as northern Litsea (*Litsea subcoriacea* Yang et P.H.Huang), short-tailed blueberry (*Vaccinium carlesii* Dunn), and red myrtle (*Syzygium buxifolium* Hook. et Arn.).

Field measurements for LAI within the study area were conducted from 30 July to 1 August 2022, during the dawn and dusk under diffuse light conditions [18]. The Li-Cor LAI2200 plant canopy analyzer (Li-COR Inc., Lincoln, NE, USA) was employed to measure the LAI (effective LAI) of 49 circular sample plots, each with a radius of 10 m, within the study site (as depicted in Figure 1b). Measurements were taken with the plot center as the reference point, specifically at the center point of the plot and at four additional points located 5 m from the center in the east, south, west, and north directions. The sensor head was maintained at a height of 2 m above the ground during measurements. Simultaneously, we employed real-time kinematic (RTK) surveying techniques using GNSS equipment (South Surveying & Mapping Galaxy 6 RTK) to capture the central coordinates of the plots. This device integrates signals from four satellite systems—GPS, GLONASS, Galileo, and BeiDou—achieving horizontal and vertical accuracies of ±8 mm and ±15 mm, respectively, and is equipped with 336 signal tracking channels. Its precise and rapid positioning ensures the accurate alignment between ground-measured data and the corresponding point cloud. The FV2200 software (Version 2.1, Li-COR Inc., Lincoln, NE, USA) was used to calculate the LAI. Given that some plots were located on steep slopes, only data from the first four rings were selected for computation to avoid any potential interference from the sloped terrain.

### 2.2. LiDAR Acquisition and Processing

In this study, aiming to enhance the accuracy of LAI estimation in mountainous forests, three flight trajectories were established, encompassing both single and multiple flight lines (Figure 2). A DJI M600 Pro drone equipped with the RIEGL miniVUX-1UAV laser scanner was employed for data collection on 30 July 2022 (Table 1). Two flight paths were set along the perpendicular and parallel directions to the mountain orientation to acquire point clouds in the study area. The point cloud densities were measured at 89 pts/m$^2$ and 78 pts/m$^2$, respectively. During aerial surveying, the flight speed was set at 8 m/s, and the off-nadir scan angle for emitted pulses was restricted to 60°. The flight altitude was 57 m above the highest point of the mountain and 179 m above the lowest ground observation point in the valley. The spacing between flight lines was set at 60 m. Consequently, the

single-flight line scan overlap rates in the mountaintop and valley regions reached 70% and 85%, respectively. To invert LAI from the point cloud obtained along the vertical mountain orientation, and based on ground-truth data, further analysis of LAI estimation errors and the response of point cloud attributes to changes in topographic factors were conducted. In order to investigate whether changing flight directions and increasing flight lines could improve LAI estimation accuracy, this study pre-processed the collected point cloud data into three LAS files, including two single-directional and one bidirectional flight lines, named vertical to the mountain's orientation (VMO), parallel to the mountain's orientation (PMO), and cross-track flight lines (CFL), respectively. To compare the accuracy of LAI estimation among the three flight lines, we performed 50% sparsity processing on the cross-track flight line point cloud, resulting in a final point cloud density of 80 pts/m², which is similar to the density of single-directional point clouds.

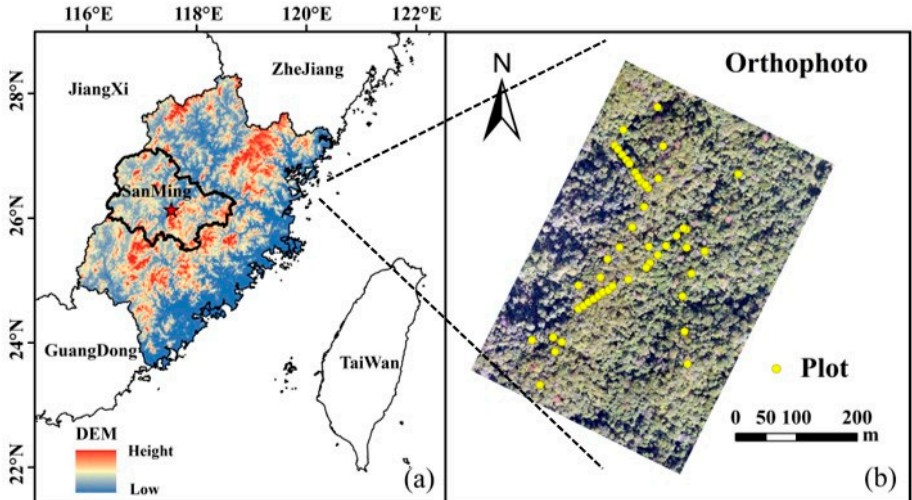

**Figure 1.** Overview map of the research area, which (**a**) showcases the location of the research area of Fujian Province, China and (**b**) shows the positions of all plots nested on an orthophoto image of the study area (WGS 84/UTM zone 50N).

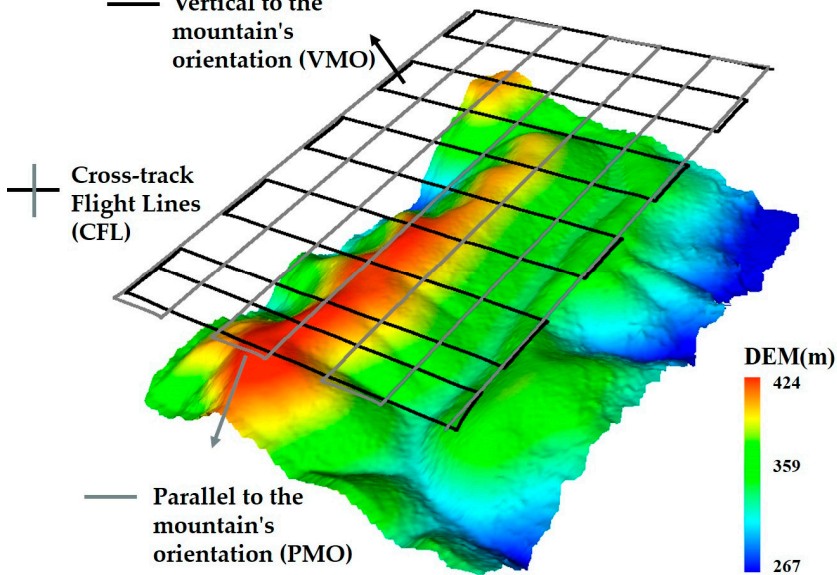

**Figure 2.** The flight trajectories utilized in the experimental flights consisted of VMO, PMO, and a composite of the two paths creating CFL. DEM height indicates the elevation data.

**Table 1.** LiDAR survey and sensor characteristics, for the RIEGL miniVUX-1UAV sensor used in this study.

| Parameters | Value |
|---|---|
| Wavelength (μm) | 1050 |
| Beam divergence (mrad) | 1.6 × 0.5 |
| 100m from the ground beam Foot (mm) | 160 × 50 |
| Pulse rate (kHz) | 100 |
| Max. Measuring Range (m) | 330 |
| Field of View (°) | 360 |
| Max. Number of Targets per Pulse | 5 |
| Accuracy (mm) | 15 |

The preprocessing of point cloud data includes noise removal, filtering, and ground point classification. Subsequently, the classified ground points were interpolated using the TIN (triangulated irregular network) algorithm to generate a digital elevation model (DEM) with a spatial resolution of 0.25 m. The first return points were then interpolated to produce a digital surface model (DSM). The canopy height model (CHM) was subsequently derived by subtracting the DEM from the DSM. Moreover, point clouds were normalized with respect to the DEM. All of these procedures were performed using LiDAR 360 (Digital Green Earth Company, Beijing, China).

*2.3. Beam Footprint and LAI Estimation*

Previous studies have shown that, the accuracy of LAI estimation is influenced by the expansion of the beam footprint when employing a point cloud simulator developed using the discrete anisotropic radiative transfer model [30]. Taking this effect into consideration, this study aims to quantify the spatial variability of the beam footprint. The diameter of the beam footprint is calculated as:

$$L = \sqrt{\frac{\left(\frac{D}{\cos(\theta)}\right)^2 \times \Omega_T^2}{4} \times 2} \tag{1}$$

$L$ represents the diameter of the beam footprint, $D$ denotes the vertical distance from the sensor to the target, $\theta$ is the average off-nadir scan angle, and $\Omega_T$ signifies the laser divergence angle.

The conventional method for estimating LAI from discrete point cloud data is grounded in the Beer–Lambert law, following the formula proposed by Richardson et al. [31]:

$$Gap = \frac{N_{ground}}{N_{total}} \tag{2}$$

$$LAI = \frac{-\cos(\theta)}{G(\theta, \alpha)} \times \ln(Gap) \tag{3}$$

where *Gap* is the gap fraction, calculated from the ratio of ground points $N_{ground}$ to the total number of points $N_{total}$ using an altitude threshold of 2 m (corresponding to the height at which the ground-truth LAI is acquired) [32,33]. $\theta$ is the angle at which the beam or ray penetrates the canopy, taken here as the average off-nadir scan angle from LiDAR; $(\theta, a)$ is the extinction coefficient, and for spherically distributed leaves, it is valued at 0.5.

In order to systematically investigate the specific impact of slope position and gradient on LiDAR discrete point cloud features (such as point density, off-nadir scan angle, and beam footprint) and LAI estimation, the study area was partitioned into three slope positions determined by elevation: upper slope (395–424 m), mid-slope (366–395 m), and lower slope (337–366 m). These categories included 20, 14, and 15 plots, respectively (refer to Figure 3). Furthermore, the plots were classified into three slope grades (0–15°, 15–30°,

and 30–41°), which resulted in the allocation of measured samples into 14, 18, and 17 plots for each grade, respectively.

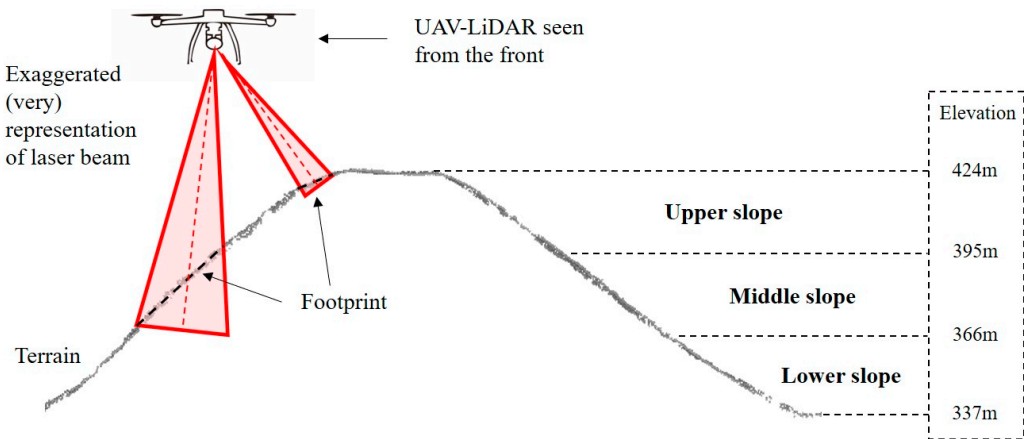

**Figure 3.** Schematic representation of slope position classification and UAV LiDAR monitoring in the study area. According to the elevation, the study area was divided into three slope positions: upper slope, middle slop, and lower slope. Due to the influence of UAV altitude and slope, there is significant spatial heterogeneity in the footprint and incident angle of laser beams reaching the ground.

### 2.4. Accuracy Assessment

We employed the Pearson correlation coefficient (R) to examine the correlation between the three point cloud attributes and the residual LAI estimation, as well as to assess how the point cloud attributes respond to changes in elevation. Additionally, we utilized the coefficient of determination ($R^2$), root mean square error (RMSE), and bias to assess the accuracy of LAI estimation. The corresponding formulas are as follows:

$$RMSE = \sqrt{\frac{\sum_{i=1}^{n}(x_i - \hat{x}_i)^2}{n}} \tag{4}$$

$$Bias = \frac{\sum_{i=1}^{n}(x_i - \hat{x}_i)}{n} \tag{5}$$

where $x_i$ is a reference observation, $\hat{x}_i$ is a predicted observation from UAV LiDAR data, and $n$ is the number of reference observations.

## 3. Results

### 3.1. Influence of Elevation on Point Cloud Feature Information and LAI Estimation

In this study, we collected point cloud data along a predefined vertical mountain route, and quantified the LAI based on the Beer—Lambert law, discrete point cloud transmittance, and the average off-nadir scan angle of the sample plots (Figure 4a). Our analysis revealed a substantial linear correlation ($R^2 = 0.64$) between the UAV LiDAR estimated LAI values and the ground-measured values. However, the UAV LiDAR estimated LAI exhibited a consistent trend of overestimation, characterized by an error represented by RMSE = 0.84 and Bias = 0.70. As elevation changed, a weak negative correlation ($R^2 = 0.07$) was observed between the ground-measured LAI and elevation, with a regression slope of −0.011. From the valley to the ridge, the forest canopy exhibited a trend of transitioning from dense to sparse. The gridded LAI data generated from the point cloud dataset reveal the vegetation structure within the study area (Figure 5a). The average LAI predominantly ranges between 4 and 6. Significantly, aside from a small section of bamboo forest in the southwest corner of the study area (Figure 5b), the DEM in Figure 2 shows that in the secondary mountain forests, LAI values less than 4 are predominantly found at the mountain peaks (Figure 5a). Meanwhile, LAI values between 6 to 8, as well as those exceeding 8, are mainly found in the

valley regions. The competitive environment for sunlight and increased nutrient utilization rates in the valley resulted in denser tree growth compared to the ridges [34]. Concurrently, the LAI values estimated using LiDAR showed a more pronounced negative correlation ($R^2 = 0.27$) with elevation, featuring a regression slope of -0.026 (Figure 4b). This indicates the influence of elevation on LiDAR parameters when estimating LAI.

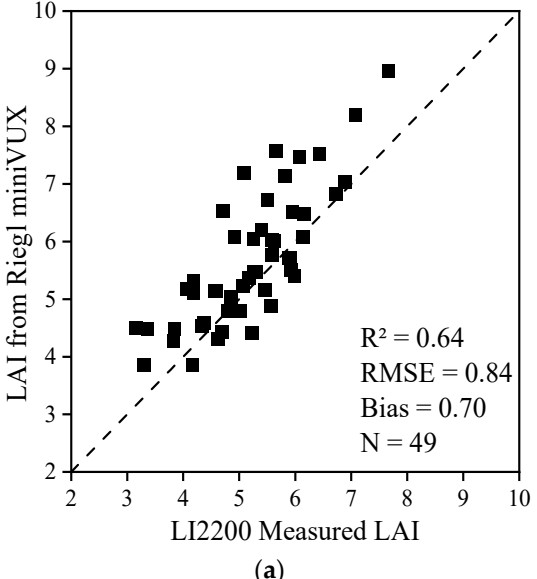
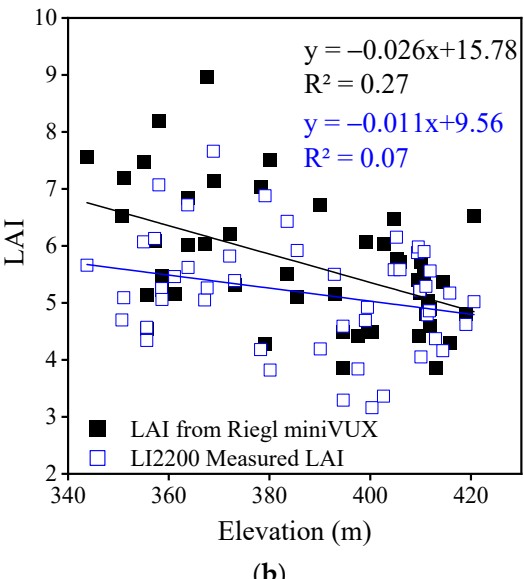

**Figure 4.** (**a**) Comparison between the inferred LAI at the plot scale and the ground-measured values; (**b**) variation of ground-measured LAI and estimated LAI with elevation.

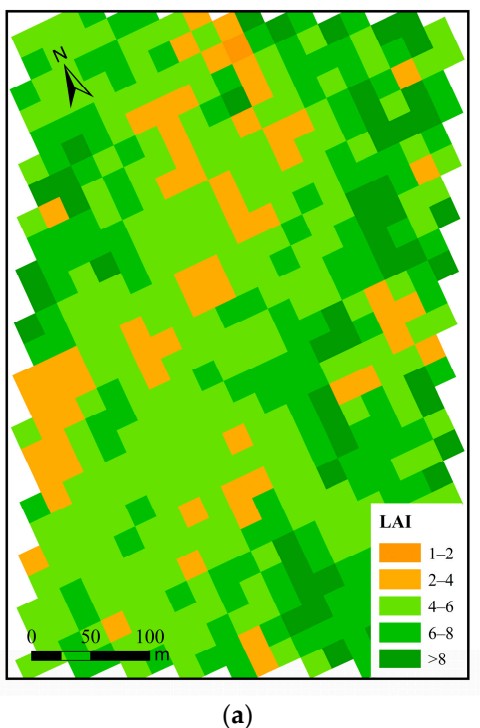
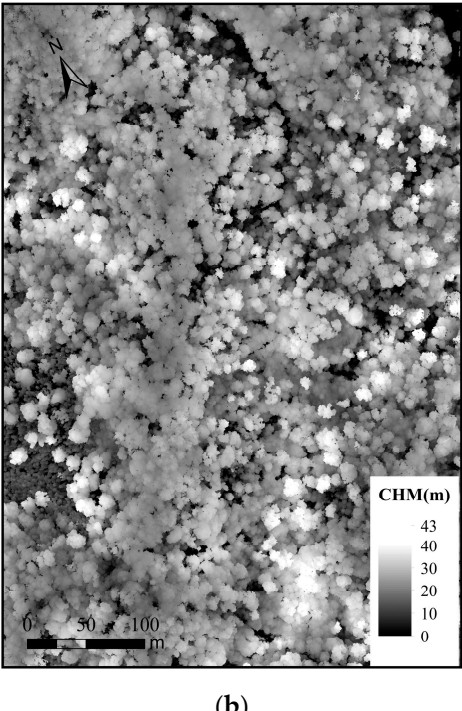

**Figure 5.** Spatial distribution maps of vegetation structure in the study area. (**a**) This figure depicts the spatial distribution of LAI within the study area, offering a visual representation of vegetation density and distribution. (**b**) This figure presents the CHM for the same study region, illustrating the vertical structure and height variations of the vegetation canopy.

Further analysis uncovered a relationship between LAI estimation residuals and point cloud feature attributes (Figure 6). The residuals in LAI estimation showed a mild negative correlation with point density (R = −0.46) and increased as the laser beam footprint expanded (R = 0.48). In mountainous study areas with constant height measurements by the LiDAR sensor, the range of laser pulses reaching lower elevations lengthened, causing the laser beam to diverge and the footprint to expand. Additionally, the average scan angle of the point cloud within the sample plots emerged as a crucial factor influencing LAI estimation, demonstrating a moderate negative correlation with estimation residuals (R = −0.47).

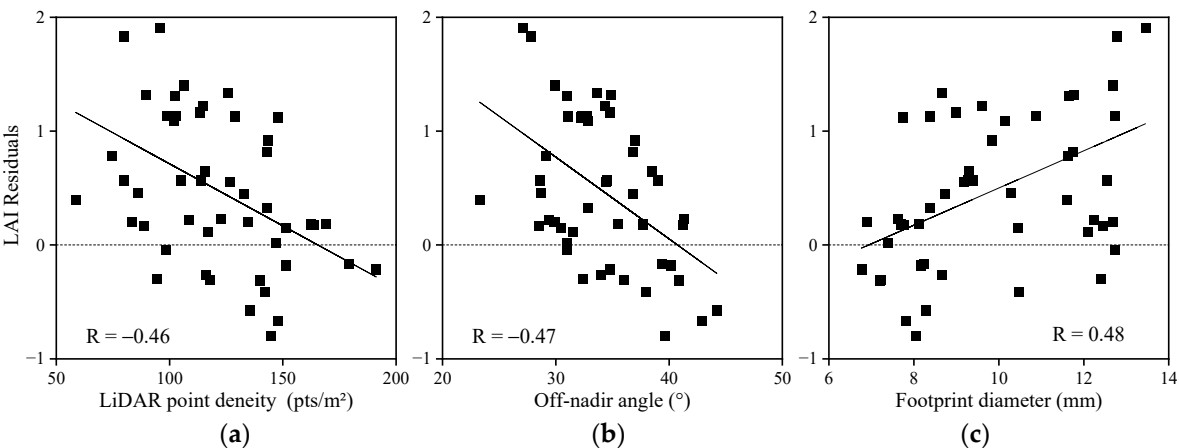

**Figure 6.** Variation of LAI residuals with point density (**a**), average off-nadir scan angle (**b**), and beam footprint (**c**).

The scatter plot depicts the changes in three point cloud features with elevation reveals that plot point density, average off-nadir scan angle, and beam footprint width all vary with elevation, and the absolute values of the correlation coefficients with elevation have all exceeded 0.7 (Figure 7). Both point cloud density and average off-nadir scan angle exhibit positive correlations with elevation, showing correlation coefficients of 0.76 and 0.70, respectively. In the downhill area, point density can be as low as approximately 50 pts/m$^2$, while in the uphill region, it can reach as high as 200 pts/m$^2$. The off-nadir angle, denoted as $\theta$, exhibits significant spatial variability across different plots, showing an increasing trend with rising elevation. Among these, beams with a small scanning angle are more likely to penetrate the canopy and reach the ground, especially enhancing the penetration rate in valley areas. However, due to the spacing between flight lines, ground point clouds with scanning angles less than 26° exhibit significant gaps between the two flight lines. Point clouds with large scanning angles, as their transmission distance increases, experience weakened penetration ability or are blocked by the mountain, concentrating more on the mountaintop. Beams greater than 30° can hardly reach the bottom of the valley. Laser beams with large scanning angles can penetrate to some extent into the interior of the canopy, especially in the forests on the mountaintop. The complex vertical structure features of secondary broad-leaved forests are thus portrayed more comprehensively (Figure 8). However, based on Beer's Law, with the increase in off-nadir scan angle, the value of cos($\theta$) decreases, resulting in a reduced estimated LAI. Conversely, it increases with the opposite trend. Additionally, there is a significant negative correlation between beam footprint and elevation (R = −0.98). In the uphill areas, the average footprint diameter is less than 6 cm, whereas in the downhill regions, it can reach up to 14 cm, representing a twofold difference. When the footprint diameter increases due to greater sensor-to-ground distance, the energy distribution area of the laser pulse also enlarges, resulting in decreased energy density. Consequently, this limitation reduces the pulse's capability to penetrate the canopy.

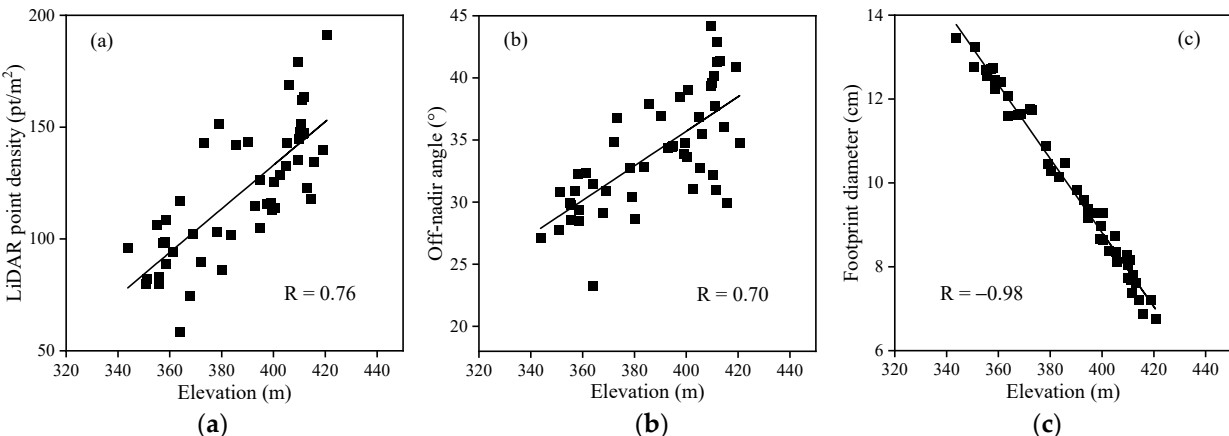

**Figure 7.** Scatter plots illustrating the variation of point cloud features with plot elevation: (**a**) point density, (**b**) average off-nadir scan angle, and (**c**) beam footprint width.

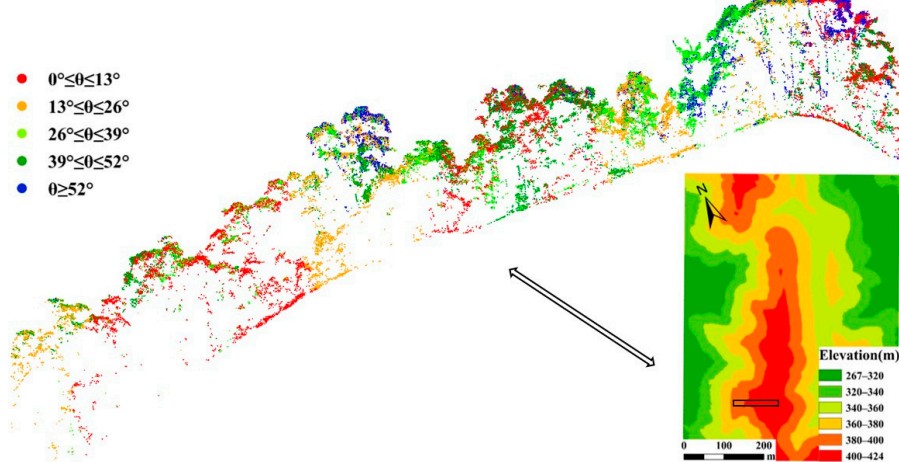

**Figure 8.** The point cloud profile after dividing the scanning angle. Vertical profile of point cloud data on the west-facing slope of the study area spanning approximately 120 m, classified by off-nadir angle categories.

*3.2. Comparison of LAI Estimation across Different Slope Positions and Slope Gradient Categories*

The influences of elevation and slope gradient distinctly affect the residual error in LAI estimation. Two-dimensional contour plots reveal an increasing trend in LAI estimation error as elevation diminishes (Figure 9). The RMSE for LAI estimation increases from 0.63 on the upper slope to 0.86 on the mid-slope, further rising to 1.07 on the lower slope. The bias also increases from 0.18 on the upper slope to 0.75 on the lower slope, indicating a significant trend of overestimation in LAI as elevation decreases (Figure 10a). At the same time, in the mid-to-lower slope regions, the absolute residual error of LAI estimation increases significantly with the increase of slope, especially in the downhill position (elevations below 366 m). For plots with a slope gradient exceeding 20° in this region, the relative error (RMSE%) in LAI estimation by LiDAR exceeds 25% (Figure 9). Additionally, as the slope gradient increases, the correlation between the observed and estimated LAI values weakens (Figure 10b). Within a 0–15° slope gradient range, the $R^2$ value reaches 0.72, with a bias of just 0.4. However, for plots within the 15–30° and 30–41° slope gradient categories, the $R^2$ values decline to 0.66 and 0.53, respectively. Notably, in the 30–41° slope category, the bias reaches a peak of 0.77.

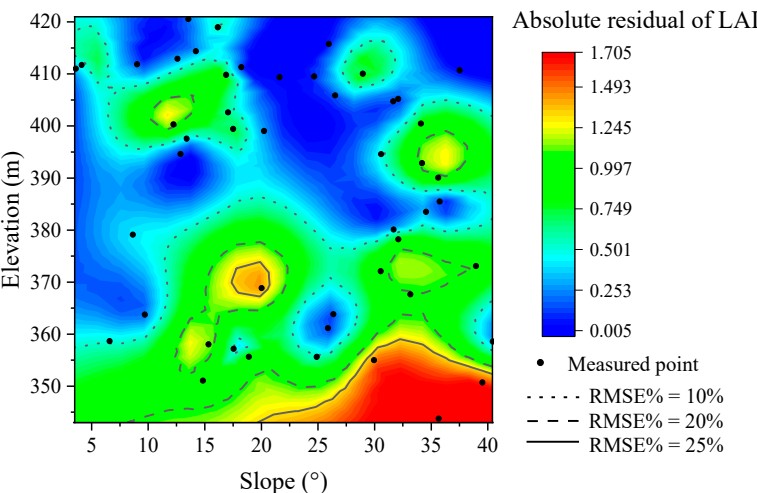

**Figure 9.** Contour plot illustrating the influence of elevation and slope gradient on LAI estimation residuals.

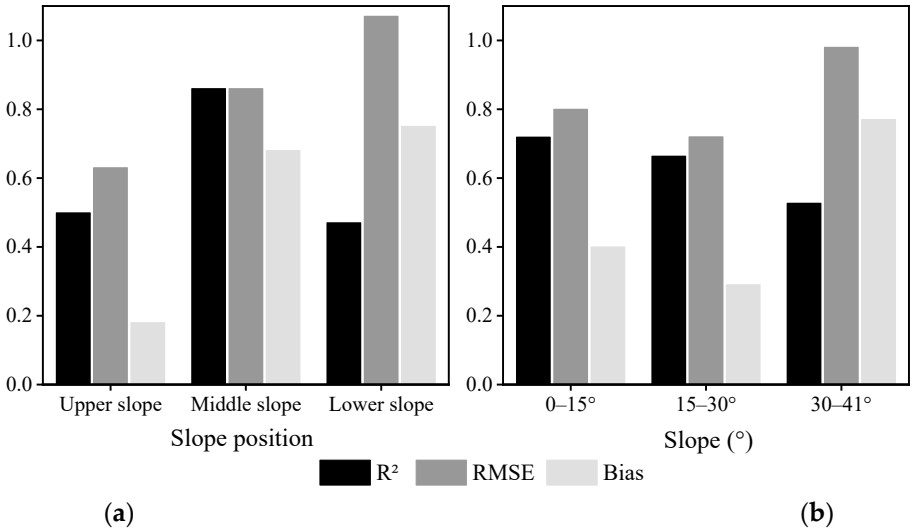

**Figure 10.** Comparison of LAI estimation accuracy across different slope positions and slope gradient classifications. (**a**) Comparison of LAI estimation accuracy across different slope positions, namely, upper slope, middle slope, and lower slope. (**b**) Comparison of LAI estimation accuracy across categorized slope gradients: 0–15°, 15–30°, and 30–41°.

*3.3. Impact of Altering Flight Directions and Increasing Flight Lines on the Accuracy of LAI Estimation*

Comparing the LAI estimation results derived from point cloud data collected during flights along vertical mountain orientations (Figure 3), the parallel mountain orientation yielded an $R^2$ of only 0.67 between the estimated and measured LAI values. The bias decreased from 0.70 to 0.44, indicating that the tendency of LiDAR to overestimate LAI appeared to be mitigated. However, there was almost no difference in the RMSE between the two single flight path estimations, with values of 0.84 and 0.85, respectively (Figure 11a). The LAI estimation accuracy noticeably improved with the cross-track dual flight line design, with an $R^2$ of 0.70 and an RMSE of 0.75 (Figure 11b). Further breaking down the results by slope position and gradient, it was observed that the $R^2$ values between the estimated and actual LAI values for different slope positions using the dual flight lines were slightly higher than those obtained using single flight lines (Figure 12). There was a significant reduction in the RMSE, with decreases of 71% and 30% for the upper and lower slope positions, respectively (Figure 12a). Among different slope gradient classifications,

the two categories with a gradient less than 30° showed increases in $R^2$—of 0.15 and 0.07, respectively—in the dual flight line estimation as compared to the single flight line results (Figure 12b). For slopes exceeding 30°, the $R^2$ between estimated and measured LAI values actually decreased when using the dual flight lines. The RMSE exhibited varying degrees of reduction. When transitioning from single to dual flight lines, the 15–30° slope gradient category showed only a minor RMSE reduction. However, for the 0–15° and 30–41° gradient categories, the RMSE decreased by 20% and 8%, respectively.

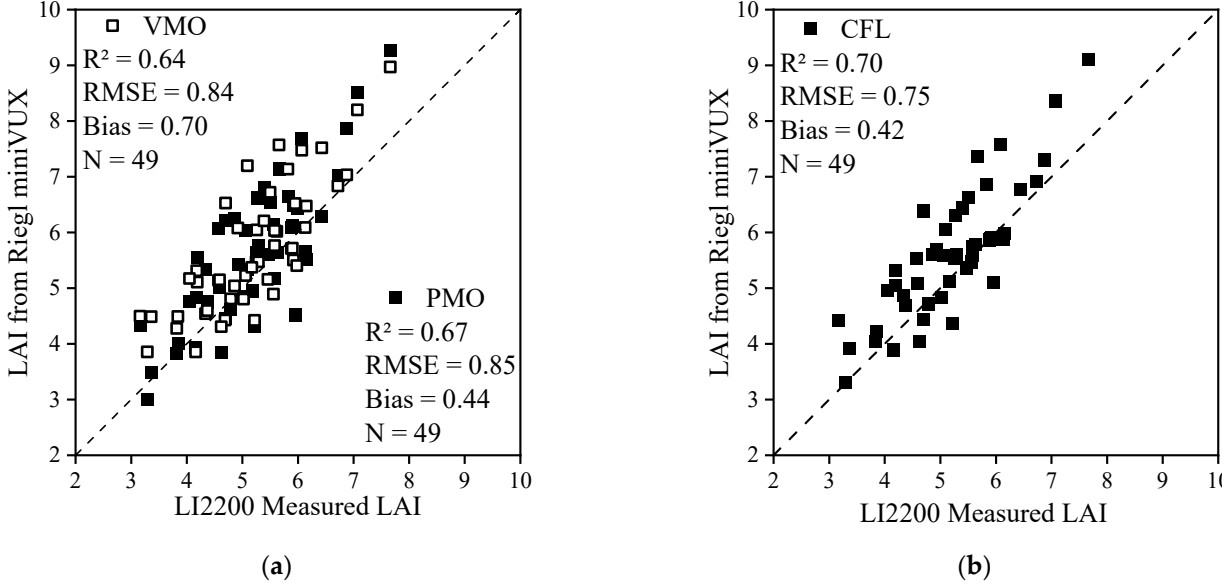

**Figure 11.** LAI estimation comparisons using different flight trajectories. (**a**) LAI estimations derived from VMO and PMO. (**b**) LAI estimation using the CFL design.

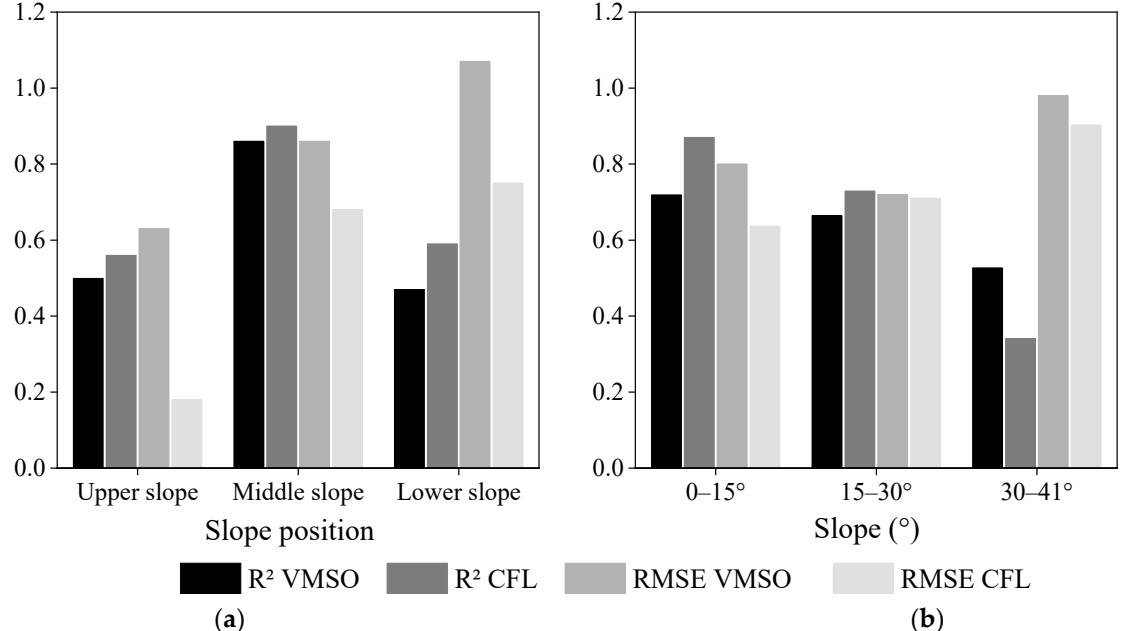

**Figure 12.** Comparison of LAI estimation accuracy for different slope positions and slope gradations between single and double flight lines. (**a**) Depicts the comparison of LAI estimation accuracy between two different flight plans (VMO and CFL) across various slope positions (upper, middle, and lower slopes). (**b**) Compares the LAI estimation accuracy of the two different flight plans (VMO and CFL) across three slope categories (0–15°, 15–30°, and 30–41°).

## 4. Discussion

### 4.1. Terrain Relief's Effect on LAI Estimation through Uneven Point Cloud Density

The complexity of terrain in mountainous forest regions, particularly the relative elevation difference, is a primary objective factor leading to significant spatial variation in point cloud density within the study area. The point cloud density in sample plots ranged between 58 and 191 pts/m$^2$, exhibiting substantial spatial variability that decreased with decreasing elevation. The main reasons for this phenomenon are as follows: (1) The energy of laser pulses decreases with the increase in detection distance. In the case of fixed-altitude flight for UAV, the energy of laser pulses reaching the canopy decreases, subsequently reducing the probability of triggering echoes. (2) Low-flying UAV laser beams with large scanning angles may be obstructed by ridges, leading to lower point cloud density in valley areas and on slopes without flight lines, resulting in low-density sample plots (Figure 13). While high point cloud density enables a detailed representation of forest canopy details, low-density point clouds may increase LAI estimation errors due to missing local canopy points. The increasing trend of LAI estimation residuals with decreasing elevation in this study supports this observation. In operations in complex mountainous terrain, UAV flight lines often fail to meet the requirements for high and uniform point density, resulting in the coexistence of high-density and low-density regions in the surveyed area [23]. However, some studies, based on the random subsampling of original point clouds classified by density, compared the impact of different point densities on canopy structure parameters. Contrary to expectations, these studies suggested that as point cloud density decreases, with minimal changes in canopy closure and gap ratio, the impact on LAI extraction is minimal. This comes at the cost of sacrificing spatial resolution in LAI extraction [35]. Even at a point cloud density of 16 pts/m$^2$, canopy structure parameters can still be extracted with 95% accuracy at a sampling scale of 5–20 m [23]. Following the detailed depiction of the forest canopy in the original high-density point cloud, sparse resampling based on random principles does not alter the pattern of canopy anisotropic characteristics. However, the existence of low-density point clouds due to terrain or flight line planning issues leads to local information gaps in the canopy and insufficient capture of directional gap ratios, resulting in an overestimation of LAI [36].

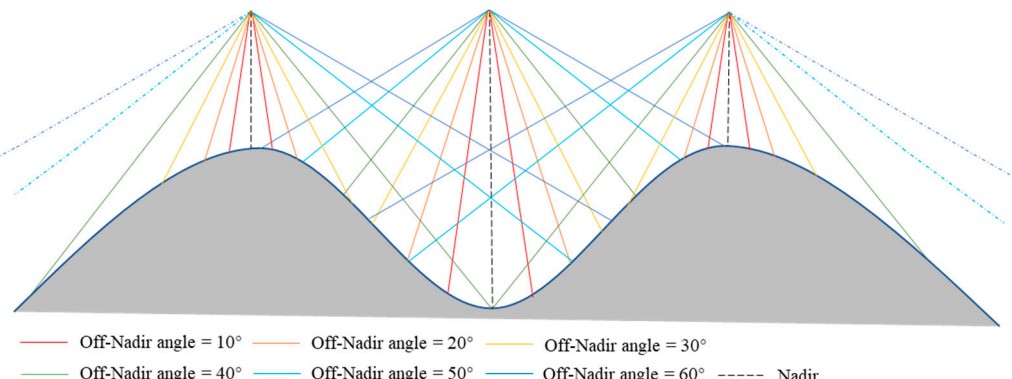

| Off-Nadir angle = 10° | Off-Nadir angle = 20° | Off-Nadir angle = 30° |
| Off-Nadir angle = 40° | Off-Nadir angle = 50° | Off-Nadir angle = 60° | ----- Nadir |

**Figure 13.** Low-flying UAV LiDAR observation mountain simulation map.

### 4.2. Impact of Terrain-Induced Beam Scan Angle and Footprint Spatial Heterogeneity on LAI Estimation

Compared to traditional airborne platforms, UAV LiDAR can achieve a higher point cloud density, primarily benefiting from not only low-altitude flight paths but also its wider scanning angle. When flight speed, altitude, overlap, and other parameters remain constant, expanding the scanning angle in the same area to achieve the specified point density allows for faster completion of measurement tasks. Alternatively, within the same time frame, a wide scanning angle can capture a larger area of point clouds. However, the efficiency advantage of wide scanning may be offset by significant biases in canopy structure parameter estimation. The data in this study represent a case in which wide

scanning angles were adopted to balance efficiency considerations. On mountain tops, more laser pulses with large scanning angles are intercepted, while in valley areas, the steep slopes on both sides obstruct the large-angle beams, making it challenging for them to reach the valley floor (Figure 13). This phenomenon is also a primary reason for the average off-nadir angle of plot point clouds reaching its maximum near ridgelines and generally being smaller in valley areas. The spatial structural variation of the average off-nadir angle with increasing elevation will result in systematic errors in LAI estimates based on Beer's Law. Meanwhile, the increased deviation of the laser beam from the nadir leads to a longer path through the canopy, heightening the canopy's occlusion effect. This tendency amplifies LiDAR's underestimation of gap fraction and overestimation of LAI, especially with wider scanning angles. Consequently, researchers have explored the impact of scan angles on estimating gap fractions using ALS data and recommended avoiding off-nadir angles exceeding 23° [27]. Brede et al. [37] showed that when the absolute off-nadir scan angle (ASA) reached 40°, the occlusion showed a difference of ≤5% compared to ASA = 0°. Subsequently, occlusion increased linearly until reaching an ASA of 60°, after which it increased significantly. Scan angles between 30° and 40° proved optimal for trunk detection [38]. Avoiding large off-nadir angles enhances gap estimation, mitigating issues in estimating gaps within heterogeneous stands characterized by between-crown gaps [37]. Zheng et al. [36] also advised against overlooking scan angle details in forest canopy parameter retrieval, especially with wide-angle ALS data (−30° to 30°). Quantifying directional gap fraction may reveal canopy anisotropy and deepen understanding of canopy radiation and reflection. The ongoing discussion on establishing the maximum LiDAR scan angle persists. A more comprehensive understanding of angle-induced biases in estimating canopy structure parameters would facilitate quantifying the balance between cost and predictive bias. Given the typically higher off-nadir angles in UAV LiDAR, methodologies should account for viewing angles during development.

The footprint size of the laser beam is another crucial instrument parameter that restricts the penetration of laser pulses through the canopy and accurately captures gap fraction and LAI. Due to the divergent nature of the laser beam, the UAV LiDAR footprint size increases with the detection distance. When collecting point clouds at a constant altitude, the footprint size increases with decreasing elevation. Similar to increasing flight height over flat terrain, the rated pulse energy is distributed over a larger footprint area. This results in a decrease in irradiance on the canopy per unit area, and the concentration of peak pulse power shows a decreasing trend [39]. For a pulse encountering partial interception, the remaining energy may be insufficient to trigger higher-order return, leading to the inability to obtain information about the interior of the canopy and the terrain, and even missing ground point clouds in local areas [35]. Clear evidence from this study indicates that the estimation error of LAI increases with decreasing elevation and reaches its maximum at the downslope position (Figure 12a). Additionally, there are significant differences in the beam divergence among different instruments. For example, the Reigl VUX-1LR laser has a beam divergence of 0.5 × 0.5 mrad, while the Reigl miniVUX-1 used in this study has a divergence of 1.6 × 0.5 mrad. At a flight altitude of 100 m, the footprint size reaches 16 cm × 5 cm. This leads to many gaps within the canopy that are too small to be detectable, which is a primary reason for the substantial overestimation of LAI in this study.

### 4.3. The Impact of Slope on LAI Estimation

This study focuses on mature, typical evergreen broad-leaved forests in the Central Asian belt, characterized by dense canopies and complex vertical structures. The average slope of 49 ground-measured plots is 23°, with an average (LAI of 5.13. The cross-track collection of point clouds resulted in an LAI estimation RMSE (root mean square error) of 0.75. As the slope grade increases, the correlation between measured and estimated LAI values decreases. Notably, in plots with a slope greater than 30° (17 in number), the RMSE rises to 0.89, RMSE% ≈ 17%, indicating a higher estimation error than the average for the

entire study area. Studies demonstrated that LAI products derived from remote sensing in high-altitude and rugged terrain have significantly lower accuracy compared to flat areas. Satellite-borne LiDAR (GLAS) also faces substantial challenges in forest LAI inversion in complex terrains, yielding better results ($R^2$ = 0.69 and RMSE = 0.33) only when the slope is less than 20° [40]. A simulation results based on the DART model also show that the LAI estimation error on slopes increases significantly as the slope angle increases [41]. Jin et al. [42] validated the accuracy of the GLASS and MODIS LAI products in the mountainous regions of Southwest China using high spatial resolution LAI with RMSEs of 1.72 and 1.75 and relative errors of −71% and −67%, respectively. A notable limitation of LAI estimation using discrete point clouds is that in the process of terrain normalization, the shape of the canopy and the position of the treetop may be systematically distorted, leading to reduced complexity in the vertical distribution of the plant area index (PAI). However, the degree of change in the PAI profile is not solely determined by the steepness and roughness of the local terrain but is a result of the interplay of local topography and the distribution of trees on the landscape surface. More complex terrain does not necessarily lead to more variability [28]. In reality, LAI2000 observations or fisheye images collected on the ground also face uncertainties due to terrain influences. A direct manifestation of the terrain effect is that the slope increases the gap fraction down-slope because the path length is reduced, while the up-slope gap strongly decreases down to the topographic mask where soil is seen. When computing azimuthally averaged gap values, down-slope and up-slope effects approximately compensate [15]. Whether terrain effect correction is necessary remains a subject of debate [14]. For dense forests like those in this study, estimating LAI based on discrete point clouds is an efficient and accurate method. Although larger slopes indeed affect the correlation between LiDAR estimates and ground measurements, the estimation error does not monotonically increase with slope grade. Future slope effect evaluations may benefit from the "incidence angle normalization" method to enhance the correlation between LiDAR and ground measurements. This involves dividing ground measurements taken parallel to the slope by the cosine of the slope angle and adjusting the LiDAR scanning angle to be relative to the normal of the slope surface, then calculating LAI based on Beer's law.

### 4.4. Optimizing Flight Planning and Prospects for LAI Estimation in Mountainous Forests

In mountainous forest survey areas, the undulating terrain leads to significant spatial heterogeneity in point density, scanning angle, and laser footprint, which substantially affects LAI estimation based on UAV LiDAR. High-power, long-range, and small-footprint airborne laser scanning instruments are preferred for estimating structural parameters in mountainous forests. Additionally, the design of UAV flight paths is crucial for the quality of the collected point cloud data. Cross-track dual flight line designs more comprehensively collect forest canopy gap rates with anisotropic characteristics, facilitating more accurate LAI estimation. This study aims to investigate the impact of terrain factors on point cloud attributes and forest LAI under conventional flight path designs. While this is limited to dense, randomly distributed canopies, it represents one of the most challenging scenarios. This is a preliminary attempt to optimize flight path design to improve the accuracy of canopy structural parameter estimation. Although adding flight paths increases the cost of data acquisition, it also enhances the representativeness of the data [36]. There is still much experimental work to be done to improve the representativeness of LAI estimates in mountainous forests through optimized flight path design. Future experiments should explore whether terrain-following flight paths can homogenize point cloud density and scanning angles, thereby increasing the robustness of LAI estimation. It is important to coordinate flight altitude, maximum scanning angle, and overlap rate to capture more comprehensive canopy characteristics, while also considering data collection efficiency. In conclusion, fieldwork in near-ground remote sensing studies should be valued. Flight survey plans should be scientifically and rationally set according to terrain and other relevant environmental factors, aiming to obtain high-quality point cloud data.

## 5. Conclusions

In this study, using UAV LiDAR technology, we explored the influence of terrain factors on the estimation of LAI in subtropical mountainous secondary broad-leaved forests. Based on field investigations and data collection in the *Castanopsis fargesii* Natural Reserve in Sanming City, Fujian Province, we drew the following conclusions.

The LiDAR estimated that LAIs from the unidirectional UAV flight path display a strong linear correlation ($R^2 = 0.64$) with ground-measured values in subtropical mountainous secondary broadleaf forests. However, the LiDAR overestimates the ground-measured values (RMSE = 0.84, Bias = 0.70), and exaggerates the trend of increasing forest LAI with decreasing altitude. Slope acts as a limiting factor in the correlation between estimated and ground-measured LAI values. While slopes > 30° exhibit higher residuals compared to the study area average, no clear monotonic relationship exists between LAI estimation error and slope changes.

Altitude governs the spatial distribution of plots' point cloud attributes (point cloud density, average off nadir angle, and average beam footprint diameter), and the LAI estimation residuals have a significant linear correlation with these attributes. As the point density and off-nadir angle increase with altitude, the LAI estimation residuals continue to decrease at first, followed by a transition in LiDAR's overestimation of LAI measured values to underestimation. Additionally, with larger beam footprint diameters, LAI overestimation becomes more obvious. The mountain obstruction leads to an increase in large scan angle pulses returned from its peak, resulting in a higher average off-nadir angle of the point cloud on the mountain top. According to the Beer–Lambert law, a lower $\cos(\theta)$ yields a smaller LAI value. This simply offsets the overestimation of LAI caused by insufficient LiDAR penetration in the dense canopy and reduces the estimation residual error of LAI in the mountaintop area.

UAV flight path design also determines the quality of the collected point cloud. Compared to a single flight path, cross-track dual flight line designs more comprehensively collect the forest canopy gaps with anisotropic characteristics. This improves the correlation between estimated and ground-measured LAI values in mountainous forest survey areas, reducing estimation errors across different slope positions and gradient levels. Further optimization in flight path design can enhance the representativeness of LiDAR-based estimation of LAI in mountainous forests. Exploring the impact of terrain-following flights, optimal scan angle ranges, and overlap settings is essential for refining LAI estimation.

**Author Contributions:** Y.L. and H.Z. conceived the study. Y.L., H.Z., J.X. and G.M. performed the analysis and wrote the initial draft of the paper. All authors have read and agreed to the published version of the manuscript.

**Funding:** This research was funded by National Key Research and Development Program of China (Grant number 2020YFA0608701), Fujian Provincial Public Welfare Research Institute Basic Research Project (Grant number 2023R1002001), Education Department Project of Fujian Province (J1-1402).

**Data Availability Statement:** The data presented in this study are available upon request from the corresponding author. They are not publicly available as they belong to a national key research and development project, for which the project team has signed a data confidentiality agreement.

**Acknowledgments:** The authors would like to express their appreciation for the support provided by the members of the SubTropical Ecosystem and Atmosphere Research team from Fujian Normal University during the field investigation of this study.

**Conflicts of Interest:** The authors declare no conflict of interest.

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
