# Peer review of "Influence of Topography on UAV LiDAR-Based LAI Estimation in Subtropical Mountainous Secondary Broadleaf Forests"

_forests, doi:10.3390/f15010017_

Round 1
Reviewer 1 Report
Comments and Suggestions for Authors
Dear author,
I have the utmost respect for the effort you've put into your article. I want to express my gratitude on my behalf for your meticulous work. Your article and research are well-prepared in their current state, but I believe that with some improvements, they can reach their true potential. I have outlined these points below.
Figure 2 appears to be somewhat small and subdued. It is believed that increasing the size of the figure along with its resolution would be beneficial.
The experimental design is thought to be well-structured in line with the study objectives.
With the flight altitude remaining constant, it is believed that investigating the relationships between terrain height, LiDAR point density, off-nadir angle, and footprint diameter would not contribute significantly to the literature (as depicted in Figure 6 and its descriptions). This is primarily due to the uncontrollable atmospheric conditions encountered by the aircraft. Additionally, the off-nadir angle is not strictly perpendicular to the flight direction; it can be termed as a "swath angle," and data is acquired at an angle dependent on the scanner's configuration, in the flight direction.
References to Figure 8, Figure 9, and Figure 10 are missing within the text.
While theoretical explanations are provided in obtaining LAI values in the study, LAI data is not presented in a mapped form. In-depth explanations of the techniques used to obtain LAI values in the study are expected. The key aspects of the study are the graphs and explanations comparing the LAI values obtained using "LI2200" and "Riegl miniVUX." However, it is believed that some variables discussed in the study are not compared in these sections. For instance, change matrices can be created for values obtained from both sensors on a slope basis. Furthermore, many variables not mentioned or explored within the article can also be included in the study. The study has the potential to contribute to the literature in its current form, but there are several aspects that can be improved. There are no significant language errors in the writing, but there are passages that may fatigue the reader. In my opinion, a study of this kind should have strong visuals. Unfortunately, non-engaging visuals have been used.
Page 2 Line 64: UAV Even though it is mentioned in the title and the reader knows what this abbreviation means, an explanation should still be written.
Page 4 Table 1: 100m from the ground beam Footprint (mm)
Page 5 Line 170: 。𝜃
In short, for each parameter in your study (such as slope, scanning angle, and height), you can create classes, for example, and assess how the data from both sensors has changed within these classes. You can also evaluate what factors influence point density. When you consider the answer to this question, you can actually envision the additional dimensions that can be incorporated into your study. The data is already available to you; it's just waiting for your utilization.
In this context, my evaluation about your paper is that it requires a major revision as there is potential for further in-depth exploration in the study.
Best regards.
Reviewer 2 Report
Comments and Suggestions for Authors
Dear Autors and Editor,
I had the pleasure of reading the article “ Influence of Topography on UAV LiDAR-Based LAI Estimation in Subtropical Mountainous Secondary Broadleaf Forests”.
Overall the article has the correct structure and is very interesting to me. The presented research is well organized, and the methodology and statistics are correct. Actually, I have no special comments on this text because it is simply good. The only thing I would do was to test the modification of flight overlap. This would be interesting to see the differences according to this factor. The overlap is the factor that in many cases vary between time/batteries and accuracy so we have to balance with that. Anyway the LiDAR authors have is my dream.
Concluding this is really good research. I wish authors further succeed in publication.
Here are some small comments you can find below.
31-85
I think you can mention a little bit about ground methods of assessing the LAI. A good example is hemispherical photographs https://www.schleppi.ch/hemisfer/ or other ground measurement methods like LAI plant canopy analyzer.
89
I’m afraid it wrong value – it should be 26°11′28″N, 117°28′10″E. Simply copy and paste this value to Maps Google. The figure 1 confirms I’m right.
113
GNSS not GPS
143
There are no cross-track flight lines
263
Correct residual word on figure 7
292
On the figure put letters A and B and describe correspond letters in the figure description.
293
Remove the underlined from the legend
Reviewer 3 Report
Comments and Suggestions for Authors
The manuscript deals with a very interesting topic, but the way it was presented in my opion cannot be appealing for a broad international journal, especially for two main flows: 1) the topic is so…common: papers about LAI started on late 90s and this paper lacks a deep reference research on the topic (I disagree with several references...), it does not clear to me the novelty of your work and the effective perspectives of its application for remote sensing users and foresters that must be stated to provide the readers with really relevant and meaningful take-home messages. 2) there are too many typos: the document seems an advanced draft more than a final work... missing spaces, wrong formatting, etc can be found everywhere and I do not think it is acceptable.
For these reasons, I do not have “Line by line suggestions” but, more general advices for the Authors that are distinctively invited to:
- summarize some parts and just focus to the main results, findings, etc.
- improve introduction and discussion in terms of references about the topic and taking into account the whole bibliography and the case studies available in the world about LAI (not only Asia), avoid generalization, simplification or obvious statement for remote sensing usrs or foresters;
- carefully check the text, English is great, but typos, missing dot, spaces, etc… are not
- conclusions: provide the readers with really relevant and meaningful take-home messages. Avoid generalization. State clearly why your work is a step forward.
Comments on the Quality of English Language
Despite several typos (missing spaces, apex, etc.) i do not find any issue with english
Round 2
Reviewer 1 Report
Comments and Suggestions for Authors
Dear Author(s),
Thank you for your second round revisions. I carefully read the article and the authors' response. I saw that almost all my concerns in the previous round were answered satisfactorily. Therefore, my opinion about the article is "Accept in present form".
Best regards.
Reviewer 3 Report
Comments and Suggestions for Authors
Although I recognize a huge effort on improving the paper (and you did it), I am still not satisfied about a couple of things, that must be solved:
1) references: it is not enough, change 10% of total, for this topic, it is a clear way to do it quickly and not deeply. The total number of quoted paper is also reduced
2) typos: the text is still full of formatting problems, and you added new ones on the part you modified (some examples: line 33 missing space; line 114 missing space, line 147 th missing apex, table 1, missing spaces within the rows, line 197 too much spaces; line 517 wrong font; line 538, missing bold; every species name has patronymic in italic, and it must not be... these are just something i have found in 5 minutes...)
3) about instruments and tools... it seems you do not the topic... so if you want to add information about such things you must insert something that remote sensing users knows... for examples... if you talk about GNSS or RTK, you must add information about how it is performed, which antenna, how many channels, connection and not the brand of the instrument..).
Comments on the Quality of English Language
English language is fine for me.
